# Social Referencing in the Domestic Horse

**DOI:** 10.3390/ani10010164

**Published:** 2020-01-18

**Authors:** Anne Schrimpf, Marie-Sophie Single, Christian Nawroth

**Affiliations:** 1Max Planck Institute for Human Cognitive and Brain Sciences, Department of Neurology, 04103 Leipzig, Germany; 2Physiology Weihenstephan, Technical University of Munich, 85354 Freising, Germany; mariesophie.single@gmx.de; 3Leibniz Institute for Farm Animal Biology, Institute of Behavioural Physiology, 18196 Dummerstorf, Germany

**Keywords:** human–horse communication, social referencing, horses, emotion recognition

## Abstract

**Simple Summary:**

Daily horse handling is associated with a risk of injury. It is not clear how much (a) handlers’ emotional expressions (happy versus anxious) or (b) breed type influence horses’ behavior in new, potentially threatening situations and thus contribute to risks. We therefore assessed how horses responded to a novel object when a human handler introduced the object either with a positive (happy) or a negative (anxious) emotional expression. We found that horses in the positive condition seek more proximity to the object compared to horses in the negative condition. Furthermore, horses in the negative condition showed more vigilance towards the object (i.e., increased number of gazes) than horses in the positive condition. Independent of condition, we found in thoroughbreds less human-directed contact (interaction and gazes) than in warmbloods and ponies. We conclude that the handlers’ visual and acoustic emotional expressions affect horses’ responses to unfamiliar situations.

**Abstract:**

Dogs and cats use human emotional information directed to an unfamiliar situation to guide their behavior, known as social referencing. It is not clear whether other domestic species show similar socio-cognitive abilities in interacting with humans. We investigated whether horses (*n* = 46) use human emotional information to adjust their behavior to a novel object and whether the behavior of horses differed depending on breed type. Horses were randomly assigned to one of two groups: an experimenter positioned in the middle of a test arena directed gaze and voice towards the novel object with either (a) a positive or (b) a negative emotional expression. The duration of subjects’ position to the experimenter and the object in the arena, frequency of gazing behavior, and physical interactions (with either object or experimenter) were analyzed. Horses in the positive condition spent more time between the experimenter and object compared to horses in the negative condition, indicating less avoidance behavior towards the object. Horses in the negative condition gazed more often towards the object than horses in the positive condition, indicating increased vigilance behavior. Breed types differed in their behavior: thoroughbreds showed less human-directed behavior than warmbloods and ponies. Our results provide evidence that horses use emotional cues from humans to guide their behavior towards novel objects.

## 1. Introduction

The domestication of *Equus caballus* has been dated back 5500 years ago. The beginning of horse domestication and, subsequently, the dramatic change in mobilization has been an important driver of human civilization [1,2]. Until today, the human–horse relationship is characterized by close interaction and partnership, which requires mutual interspecies communication and learning, including expressions such as posture, vocal signals, gazes, or emotional cues [3,4]. However, it is less clear how horses respond to human emotional cues in situations of uncertainty. The knowledge about horses’ ability to adjust their behavior towards the environment by using human emotional cues has the potential to improve the horse-human relationship.

Social referencing is the ability to use others’ affective display in an ambiguous situation to form one’s own evaluation of that situation and adjust behavior accordingly [5,6]. Social referencing usually includes both referential gazes towards the sender and behavioral adjustment based on the received content of the message [7]. Walden and Ogan [6] proposed that, in humans, social referencing develops at the end of the first year of life. In their study, children between 6 and 22 months were presented with two toys simultaneously that were introduced by the parent, one with a happy and one with a fearful message. As predicted, children spent less time interacting with the toy associated with the fearful message, indicating that the parents’ affective message guided their behavior. A neuroscientific approach suggests that the foundation for social referencing in humans develops much earlier, showing that in three-month-old infants, the neural responses to objects differed according to an adult’s emotional expression and showed enhanced activity towards objects that have been negatively cued [8].

It has been hypothesized that—due to the selection of the most tame and tolerant individuals—the process of domestication increased animals’ ability to effectively read human cues and, conversely, communicate with humans [9,10]. Dogs are particularly sensitive to human social cues [11], especially human communicative pointing gestures [12] and attentional cues (such as body and head orientation) [13]. Studies also suggest advanced skills in social evaluations, as dogs preferred food from a person that previously had not been seen interacting negatively with their caregiver [14]. Similarly, dogs preferred receiving food from a person who had previously shared food with another person while they watched, rather than from a person who was observed withholding food [15]. Moreover, the owners’ response towards a stranger (approach, stand still, or retreat) has been found to influence dogs’ responses, resulting in referential looks in all conditions as well as in more interactions with the owner and less interactions with the stranger in the retreat condition [16].

Research on social referencing in dogs suggests that they can regulate their behavior according to the emotional message of the caregiver: dogs were first presented with an unfamiliar object and, in a next step, were given either a happy or fearful emotional message from their owners. In the first step, most of the dogs showed alternating gazes between the novel object and their owners, indicating that the dogs were searching for further information about the uncertain situation from the owner. In the second step, when provided with an emotional message, dogs in the negative condition showed more static behavior and inhibited movements compared to the positive condition [17]. Further, when dogs observed different emotional messages (happy vs. neutral, happy vs. disgust) towards two boxes containing food, they discriminated between the emotional messages of happiness and disgust, but not between the happy and neutral message, and preferred the box presented with a happy compared to a disgusted emotional message [18]. However, the emotional message in the letter study was provided by an unfamiliar human. It has been shown that familiarity might play an important role: If the message was delivered by the owner compared to a stranger, dogs distinguished well between both happy vs. fearful and happy vs. neutral but less well between fearful vs. neutral messages in a choice task [19]. Similarly, Yong and Ruffman [20] found that dogs are more attentive to fearful and neutral than happy emotional expressions when confronted with an ambiguous object, indicating that dogs might have difficulties interpreting negative and neutral human emotional cues.

Compared to dogs, cats are less likely to use attention-getting behavior to indicate the location of a hidden food to humans [21]. However, cats perform as well as dogs in reading human pointing gestures in an object-choice task [21]. Cats also have been found to interact more with their owners (but not with unfamiliar humans), when their owners had happy facial expression compared to a negative expression [22]. A study on social referencing showed that cats’ frequency of gaze alternations and interactions with owners was higher when the owners expressed a negative compared to a positive emotion towards a novel object [23]. In conclusion, dogs and cats understand, at least partially, human emotional messages and adjust their behavior accordingly. However, to date, only domestic companion animals have been investigated for their social referencing abilities, and it is not clear if other domestic animals show similar capacities.

Horses are particularly interesting as they have not only been selected for meat, easy temper, or performance, but also for being a human-oriented and cooperative partner [2]. Today, horses are mainly kept for recreational or sports purposes, and most recreational riders seek companionship with their horses [3]. Consequently, it can be expected that horses show advanced communicative skills towards humans. Indeed, horses, as dogs, interpret subtle human head and body cues and showed differing behavior depending on whether a human is paying attention to them or not [24,25]. They also prefer to approach or obey an attentive rather than inattentive experimenter [26,27]. In an object choice task, horses utilized human-given pointing gestures [28,29] and were equally proficient at utilizing auditory or combined visual and auditory cues to choose correctly [30] but were not able to use other cues such as body orientation or gaze [31]. Horses can also communicate knowledge of a hidden, inaccessible food to a naïve human by using visual/indicative and tactile signals [32,33]. As in dogs and cats, familiarity to the caretaker enhanced horses’ skills at interpreting human cues [24,27,34]. Studies suggest that horses differentiate between happy and angry human facial expressions [35] (but also see [36]), prefer approaching a person with a submissive rather than a dominant body posture [37], and express more vigilance behavior in the presence of negatively-valenced compared to positively-valenced human voices [38]. Familiarity of the caretaker might increase horses’ abilities to differentiate human emotional states [39]. These studies indicate that horses use facial, vocal, and postural signals to interpret human emotional states. However, it is not clear if horses would use the obtained human emotional information and change their behavior in an uncertain situation accordingly. To the best of our knowledge, social referencing from humans has not been studied before in this species.

Further, as different horse breeds have been selected for different purposes [40], their communicative skills might vary between breeds. For example, some breeds, such as thoroughbreds, were mainly selected for their endurance [41], whereas others, such as most of the warmblood horses, were also selected for being cooperative and easy to handle [42]. Previous studies suggest that breed type might play a role in human–horse communication, showing that ponies performed better than horses in an object choice task when the experimenter used visual compared to auditory cues [30] and that horses sought more contact with humans than ponies when confronted with an unsolvable task [43]. Further, with increasing percentage of thoroughbred blood lines in warmblood horses, cooperation with the rider driven by a fear response decreased in a novel object situation [44]. However, studies investigating breed type differences in human–horse-interactions are sparse.

In sum, domestication supposedly adapted companion animals, such as dogs and cats, to use human emotional cues in situations of uncertainty. To investigate whether this also extends to horses, we examined horses’ ability for social referencing from humans. We used a modified test setup previously applied to companion animals [17,23,45]. Test subjects were initially exposed to an unfamiliar stimulus, while an experimenter was constantly stating an either positive or negative expression using vocalizations and facial expressions. We assessed whether horses differentiate between positive and negative emotional content and whether these differences indicate that they comprehend the underlying emotional content. Horses’ behaviors, such as gazing behavior, position in test arena, physical interaction with humans, and physical interaction with objects, were analyzed. If the emotional expression provided by the experimenter is influencing horses’ behavior towards a novel object, we expected to find differences in their behavior (gaze, position, interaction with human and object) between the negative and the positive condition (Hypothesis 1). Specifically, as in previous studies [17,23,45], we expected to find more interaction with and/or proximity to the object in the positive condition and more interaction with and/or proximity to the experimenter in the negative condition. We further hypothesized, in line with the literature [30,41,42,43,44], that different horse breeds will behave differently when confronted with a novel object (Hypothesis 2), that is we expected breeds that have been more strongly selected for cooperative behavior (e.g., warmbloods, ponies) to seek more information from the human by interacting more with and gazing more towards human than breeds that have been more strongly selected for performance (e.g., thoroughbreds). Lastly, some studies suggest sex differences in horses’ behavior during training or handling [46,47]. We therefore examined potential sex differences in horses’ behavior during the social referencing paradigm.

## 2. Methods

### 2.1. Participants

Horses were tested at an equestrian ranch in Germany (Reiterhof Pissen, Leuna, Germany) from July 2016 to October 2017. The ranch keeps approximately 70 horses of varying breeds in large social groups, with daily access to paddocks and pasture, and with *ad libitum* hay access. Inclusion criteria included being accustomed to human handling as well as a minimum height (above 1,20 m shoulder height) to ensure that the horses could easily spot the head movements of the experimenter. A total of 47 individuals were recruited, of which 46 were included in the analysis (age ranged from 1 to 26 years; 18 mares, 25 geldings, 3 stallions, 7 ponies, 19 thoroughbreds, 20 warmbloods, Table 1). One subject was excluded due to record failure.

As we recruited subjects of varying breeds, we further classified them into the category pony, thoroughbred, and warmblood according to the following definition: Pony included all breeds with an adult shoulder height of below 1.48 m, determined in accordance with the standard pony definition of the International Federation of Equestrian Sports (FEI, Veterinary Regulations 2019). The category thoroughbred defined all horses with >75% thoroughbred blood lines in their pedigree (based on owners’ reports). Warmblood was defined as horses with <75 thoroughbred blood lines in their pedigrees (based on owners’ reports).

### 2.2. Ethics Statement

Animal care and experimental procedures were in accordance with the guidelines for the treatment of animals in behavioral research and teaching by the Association for the Study of Animal Behavior [48] and the German animal welfare legislation. The procedure used was approved by the regional veterinary control board.

Informed consent was obtained from all horse owners: Subjects were recruited via announcement in the horse owners changing room, where the owners gave their consent prior to the study by entering their horses name, age, sex, and breed in an appended list. The announcement stated information about the procedure and the publication of anonymous data. All horse owners offered their horses’ participation in the experiment of their own free will.

Daily experimental procedures took place in a familiar environment and lasted no more than 10 min per horse. The tests did not cause the horses any pain, suffering, or damage and would have been terminated if a horse had shown signs of stress (e.g., increased alertness, locomotion, or vocalization).

### 2.3. Procedure

Subjects were tested individually by the experimenter and in absence of the horse owner but with olfactory and acoustic contact to group members. At the beginning of the test trial, the experimenter led the subject in the middle of a round test arena (diameter: 18 m) and removed the halter when both were facing a novel object (a blue plastic bin covered with a blue-yellow shower curtain; height: 120 cm, diameter: 70 cm) placed opposite the entrance and located at approximately 7 m distance from the subject and experimenter (Figure 1). Horses were randomly assigned to either a positive or a negative treatment and received one single test trial. During the test trial, the experimenter alternated her gaze and voice between the novel object and the test subject. In the positive treatment (*n* = 22), she transferred a positive emotional expression: excited facial expression, relaxed body posture, and positive vocalization (‘This is great’), repeated every 10 s. In the negative treatment (*n* = 24), the experimenter transferred a negative emotional expression: anxious facial expression, tense body posture, and negative vocalization (‘This is terrifying’), repeated every 10 s. The test trial duration lasted between 60 and 120 s. At the end of the experiment, the subject was approached with praise, haltered, and brought back to the stable by the experimenter.

### 2.4. Data Scoring and Analyses

Each trial was video recorded by a wide-angle camera (GoPro Hero), placed on a tripod outside of the arena. The trial started when the halter was pulled down and ended with the catching of the horse. Each subjects’ behavior was scored using BORIS [49], including frequency of gazing behavior and physical interactions (towards/with either object or experimenter) as well as duration of subjects’ relative position to the experimenter and the object in the arena (behind experimenter, abreast to experimenter, in front of experimenter, Table 2). Due to varying trial duration, all measures were analyzed relative to trial duration: For frequencies (gazes, physical interactions), we calculated the number of horses’ behavior per minute, and for duration (position in arena), we calculated the percentage of total trial time.

All statistical analyses were carried out using IBM SPSS Statistics 24 (Armonk, NY, USA) with a two-sided α-level of 0.05. As age did not correlate with any behavioral measure and was counterbalanced between groups, age was not included in the analysis. Since behavioral measures were not normally distributed even after applying appropriate transformations, we used non-parametric statistical tests (Mann-Whitney or Kruskal-Wallis U-tests, respectively). Effect sizes (Cohen’s *d*) for each test with significant results were calculated. To test the first hypothesis, behavioral measures were compared between the positive and the negative condition using Mann-Whitney U-tests. To test the second hypothesis, behavioral measures were compared between the breeds (pony, thoroughbred, warmblood) using Kruskal-Wallis U-tests. Dunn’s post hoc tests with Bonferroni correction for multiple comparisons followed each Kruskal-Wallis test with a significant result. Sex differences in behavioral measures were examined using Mann-Whitney U-tests.

## 3. Results

### 3.1. Hypothesis 1: Behavioral Differences between the Positive and the Negative Condition

Gazing behavior: Frequency of human-directed gaze did not differ between conditions (*U* = 243, *p* = 0.636). However, frequency of object-directed gaze differed between conditions (*U* = 132, *p* = 0.004, *d* = 0.93), indicating that horses gazed more often towards the object in the negative (gazes per minute: *Mdn* = 3.93) compared to the positive condition (gazes per minute: *Mdn* = 1.99, Figure 2a).

Physical interactions: Frequency of interactions with humans (*U* = 231, *p* = 0.447) as well as frequency of interactions with objects (*U* = 241, *p* = 0.483) did not differ between conditions.

Relative position to the experimenter and the object: There were no differences between conditions in the subjects’ position, either behind the experimenter (*U* = 195, *p* = 0.120) or next to the experimenter (*U* = 258, *p* = 0.884). However, the time horses stood in front of the experimenter tended to differ between conditions (*U* = 180, *p* = 0.050, *d* = 0.67), indicating that horses had a tendency to stay longer in the area between the experimenter and object in the positive (percentage of total trial time: *Mdn* = 0.22) compared to the negative condition (percentage of total trial time: *Mdn* = 0.00, Figure 2b).

### 3.2. Hypothesis 2: Behavioral Differences between Breeds

Gazing behavior: Frequency of human-directed gaze differed significantly between breeds (*H*(2) = 12.00, *p* = 0.002, *d* = 1.10). Dunn’s pairwise tests indicated that thoroughbreds displayed fewer gazes towards humans compared to ponies (*z* = 2.699, *p* = 0.021, gazes per minute thoroughbreds: *Mdn* = 1.28, ponies: *Mdn* = 2.15) and warmbloods (*z* = −3.020, *p* = 0.008, gazes per minute warmbloods: *Mdn* = 2.50, Figure 3a). Importantly, frequency of object-directed gaze did not differ between breeds (*H*(2) = 4.66, *p* = 0.098).

Physical interactions: In line with gazing behavior, the frequency of interactions with humans differed significantly between breeds (*H*(2) = 13.07, *p* < 0.001, *d* = 1.18). Dunn’s pairwise tests indicated that thoroughbreds interacted less frequently with humans compared to ponies (*z* = 3.164, *p* = 0.005, interactions per minute thoroughbreds: *Mdn* = 0.00, ponies: *Mdn* = 1.57) and warmbloods (*z* = −2.799, *p* = 0.015, interactions per minute: *Mdn* = 0.93, Figure 3b). Frequency of interactions with the object did not differ between breeds (*H*(2) = 0.27, *p* = 0.873).

Relative position to the experimenter: The analysis of subjects’ position in the arena showed no differences between breeds for position either behind (*H*(2) = 0.60, *p* = 0.742), next to (*H*(2) = 1.55, *p* = 0.460), or in front of (*H*(2) = 1.87, *p* = 0.392) the experimenter.

### 3.3. Behavioral Differences between Sexes

Gazing behavior: Frequency of human-directed gaze (*U* = 244, *p* = 0.848) as well as frequency of object-directed gaze (*U* = 237, *p* = 0.735) did not differ between sexes.

Physical interactions: Frequency of interactions with the human (*U* = 219, *p* = 0.437) as well as frequency of interactions with the object (*U* = 208, *p* = 0.170) did not differ between sexes.

Relative position to the experimenter: The analysis of subjects’ position in the arena showed no differences between sexes for the position next to the experimenter (*U* = 191, *p* = 0.162). However, the time horses stood behind the experimenter differed between sexes (*U* = 148, *p* = 0.017, *d* = 0.80), indicating that female horses stayed longer behind the experimenter (percentage of total trial time: *Mdn* = 0.97) compared to male horses (percentage of total trial time: *Mdn* = 0.35). Further, the time horses stood in front of the experimenter also differed between sexes (*U* = 146, *p* = 0.011, *d* = 0.81), indicating that male horses stayed longer in the area between the experimenter and the object (percentage of total trial time: *Mdn* = 0.29) compared to female horses (percentage of total trial time: *Mdn* = 0.00).

## 4. Discussion

The present study investigated the ability of horses to adjust their behavior towards an unfamiliar object using human emotional cues. We introduced a novel object to 46 horses pseudo-randomly either with a positive or a negative emotional expression and observed their position in the arena, as well as gazing towards and physical interactions with the human and the object. We found differing behavioral responses in horses between the positive and negative condition. We also analyzed horses’ behavior depending on breed and sex and observed differences that might be explained by their different selection criteria or sex specific behavior. Our results provide evidence that in addition to companion animals such as dogs and cats, other domesticated animals are also able to adapt their behavior using human emotional cues, although the exact mechanisms that led to this behavioral change need to be further evaluated.

**Behavioral differences between the positive and negative condition.** In line with the social referencing literature on dogs and cats [17,23], we expected to find behavioral differences between positive and negative emotional expressions towards an unfamiliar object. Indeed, we found that horses that received a positive emotional expression spent more time between the object and experimenter compared to horses that received a negative expression. Conversely, horses in the negative condition gazed more often towards the object compared to horses in the positive condition. However, no differences in other behavioral parameters, such as gazing toward the experimenter or physical interactions with experimenter/object, were observed.

As in human infants [7,50], the positive emotional expression increased the approach behavior to the object in horses, whereas in dogs and cats, this effect was absent [17,23]. However, this effect might be due to the study design [45]: We provided horses with the emotional expression as soon as they were introduced to the novel object (similar to human infant studies, [7,50]), whereas dogs and cats in Merola and colleagues’ studies [17,23] were first presented with the unfamiliar object without provision of additional information. Only after 15–25 s was the emotional message provided. The authors suggested that the delayed provisioning of information may have conveyed a mixed message about the ambiguous objects’ value, which is supported by the results of the present study as well as by another study on dogs by Merola and colleagues [45], using a similar test design as the studies on social referencing in human infants. Here, dogs did show an increased approach to the object in the positive compared to the negative condition [45]. Our results indicate that in horses, as in human infants and dogs, human emotional information influences explorative behavior in a novel situation.

We further found that horses in the negative condition gazed more often towards the object, indicating a higher vigilance or alertness in horses receiving a negative emotional expression. As a social prey species, this reaction might be explained by horses’ species-typical behavior during potential threats, the so-called fight-or-flight response, that prepares the horse to respond adaptively to threats [51]. Indeed, experimental research showed that horses’ fear responses to novel objects are characterized by a chronological order of behaviors, often starting with object evaluation and ending with a flight response [52]. In line with this, horses expressed more vigilance behavior in the presence of negatively-valenced compared to positively-valenced human voices [38] as well as in the presence of novel visual and auditory but not novel olfactory stimuli [53]. We therefore conclude that the negative expression in our study increased vigilance behavior in horses.

Horses’ abilities to guide their behavior according to the human emotional expression indicate their high responsiveness to human cues. This adds to the literature showing their sensitivity to human attentive states [24,25,26,27], pointing gestures [28,29], or auditory cues [30]. A recent line of research also showed horses’ abilities to discriminate between human emotional facial and vocal expressions [35,38] and to adjust behavior such as looking time and gaze following according to human facial emotional expressions (happy, neutral, disgust) [54]. This might be explained by the observation that unridden horses react to humans more similarly as they do to conspecifics rather than to predators [55] and, thus, seem to consider humans’ communicative cues, especially during the evaluation of an ambiguous situation. However, it is not clear yet if domestication favored a predisposition of these abilities in horses. There is evidence for differences between modern domesticated and ancient horse breeds/Przewalski’s horses in genes associated with agreeableness, that are potentially relevant for interactions with humans [2]. On the other hand, experimental research suggests that horses’ sensitivity to human cues is not innate but requires experience to develop: As in human infants before the first year of age [6], horses younger than three years of age are less sensitive to human communicative cues compared to adult horses [31]. All subjects in our study were used to daily human handling, and their age did not correlate with behavioral measures. Future research should include handled and unhandled horses to compare their sensitivity to human emotional expressions in an ambiguous situation.

**Behavioral differences between breeds**. We also hypothesized that we would find behavioral differences in different horse breed types since they have been selected for varying purposes. We found that breed types differed in their behavior when confronted with a novel object, specifically in their gazing towards and interacting with the experimenter but not with the object. Whereas there were no significant differences between warmbloods and ponies, thoroughbreds gazed less often to and also interacted less often with the experimenter compared to warmbloods and ponies. Our results are in contrast to previous research, showing behavioral differences between ponies and horses [30,43]. A possible explanation is that both studies differentiated between ponies and full-sized horses but did not further differentiate full-sized horses into warmbloods and thoroughbreds. As warmbloods and thoroughbreds have been bred for different purposes [41,42], in future studies it might be advisable to differentiate full-sized horses. Our results indicate that thoroughbreds are less focused on humans when presented with a novel environmental stimulus than other full-sized breeds or ponies. Similarly, a previous study showed that thoroughbreds, compared to warmbloods, less often directed attention towards an experimenter that appeared suddenly in their box [56]. Research on breed specific personality in horses showed that thoroughbreds score within the highest on traits such as anxiousness, excitability, and dominance, whereas these traits are less pronounced in ponies or warmbloods [40,57,58]. Moreover, the percentage of thoroughbred blood lines in warmblood pedigrees increased physiological reactivity [44], indicating a genetic influence [59]. Although there is to our knowledge no study showing a direct link between breed and human contact-seeking, horses independent of breed that scored high on traits like excitability and anxiousness seek less contact with humans and show less human directed behavior such as gazes [43]. Further, more relaxed horses showed less avoidance behavior towards humans [60]. Interestingly, it has been shown that higher reactivity in horses increased their sports performance [61]. As thoroughbreds have been bred for performance in racing and endurance, high reactivity enables quick responses and might have been a performance advantage [40,41,62].

**Behavioral differences between sexes**. Finally, we examined potential sex differences in behavioral variables. We found that sexes differed in their relative position to the experimenter; namely, female horses spent more time behind the experimenter as compared to males, whereas male horses stayed longer in the area between the experimenter and the object as compared to females. These results are in line with previous research: mares have been found to be more suspicious and anxious than geldings [63], whereas geldings might be more easily desensitized during training than mares [46]. Mares have also been found to be less playful and curious than geldings [58]. Further, compared to mares, stallions physiological stress responses were less pronounced [64]. However, other studies did not find [24,65,66,67] or found opposing sex-related differences in horses’ behavior [47]. Potential sex-differences in horses’ behavioral patterns might be explained by male and female sexual hormones that are influencing the endocrine system [68]. For example, mares’ exploration of novel objects was influenced by their oestrous-cycle stage [68]. Future studies should assess horses’ sex steroids in social referencing paradigms to further interpret behavioral differences between the sexes.

**Limitations.** The experimental design has certain limitations. By using only one unfamiliar object, we cannot rule out the possibility that the higher approach rate for the object was caused by a general increase in arousal when hearing a positive emotional expression. Horses that participated in our study had daily food-related interactions with humans and heard expressions in similar positive phonetic attributes that are likely to be associated with food. Subjects, thus, might have anticipated food delivery during the expression of the positive expression, which caused them to approach and explore the object. However, human-directed interaction or gazes did not differ between conditions, which would more strongly indicate an association with the delivery of food in the positive condition. Furthermore, in our test setup, the experimenter was delivering a continuous emotional expression, only interrupted during times of object interaction. This might have avoided a positive feedback loop. For future studies in horses, using a choice task might be the next step to investigate how the emotional valence of human expressions directs their behavior (see for dogs [19]).

Furthermore, by only using positive or negative emotions, we cannot conclude whether horses preferred to approach the object because of the positive expression or avoided the object because of the negative expression. Research in dogs showed that they were more likely to discriminate emotional messages of happiness and disgust than of happiness and neutrality [18]. However, if the message was delivered by the owner, dogs distinguished better between both happy vs. fearful and happy vs. neutral but less well between fearful vs. neutral messages in a choice task. Dogs might have learned to associate their owners’ positive emotional messages with positive outcomes [19]. In horses, it has been shown that behavior or physiological stress responses did not differ between a familiar or unfamiliar handler, indicating that the human–horse-bond might be less salient in novel, potentially threatening situations compared to the human–dog-bond [69]. Future studies of social referencing paradigms in horses should, therefore, include both a neutral condition as well as familiar and unfamiliar informants.

Additionally, the horses might have relied on other cues, such as heart rate or tone pitch, that could have been directly affected by the provided human emotional expression. Future studies could, for instance, monitor the heart rate of the experimenter and the test subject to gain more precise insights into which human bodily cues most strongly affect horses’ behavior.

Lastly, we classified different breeds in the categories pony, thoroughbred, and warmblood. However, some of the breeds summarized in these categories may have differing breeding objectives and reactivity. To further evaluate breed differences in social referencing, we recommend testing homogenous horse populations.

**Implications.** Studying social referencing in horses is especially relevant for the daily management of horses to both inform caregivers about the influence of human emotional expressions on horses’ behavior and to improve horses’ welfare [55,70]. Horses’ ability to infer from human emotional expressions and react accordingly could considerably facilitate daily horse handling but has been rarely investigated. Knowledge about potential heterospecific information transfer from handlers to horses, thus, helps to improve horses’ welfare as well as human safety in the long term, especially during training or husbandry practices. New situations or objects should, therefore, be introduced to horses by a calm and non-anxious handler to support a positive evaluation of the situation and approach behavior by the horse.

As horse breed types differ in their behavior in new situations, results of this study suggest the importance of considering breed type during daily management of horses. Future studies on social behavior in horses should further investigate breed-related differences in horses.

## 5. Conclusions

We found evidence that horses use human emotional information to guide their behavior towards an unfamiliar object, indicating highly specialized socio-cognitive abilities and interspecies communicative skills in horses. Our results emphasize that the caregivers’ emotional expression influences the behavior of horses in novel situations and might be of interest for training and husbandry practices. Our results also support previous research, showing breed-related differences in behavioral responses towards novel objects in horses. However, the mechanisms leading to higher exploration rates need to be further evaluated.

## Figures and Tables

**Figure 1 animals-10-00164-f001:**
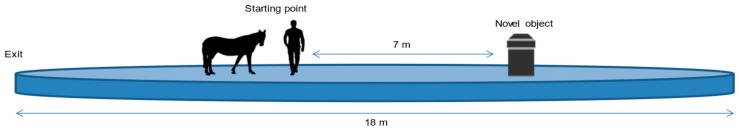
Experimental setup of the test arena as well as the subject’s and the experimenter’s location at the beginning of the trial.

**Figure 2 animals-10-00164-f002:**
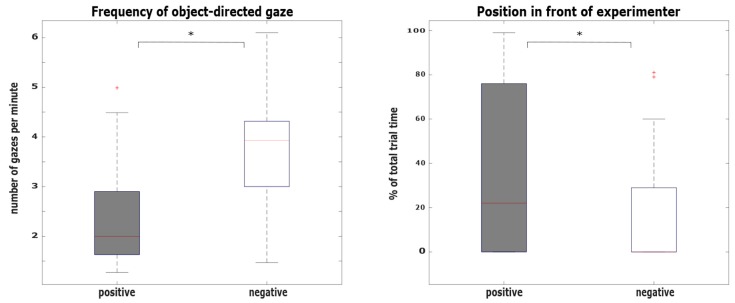
Behavioral differences between the positive and the negative condition. (**a**) Differences in frequency of object-directed gaze between conditions (*U* = 132, *p* = 0.004, *d* = 0.93). Horses gazed more often towards the object in the negative compared to the positive condition. (**b**) Differences in horses’ relative position in front of the experimenter between conditions (*U* = 180, *p* = 0.050, *d* = 0.67). Horses stayed longer in the area between experimenter and object in the positive compared to the negative condition.

**Figure 3 animals-10-00164-f003:**
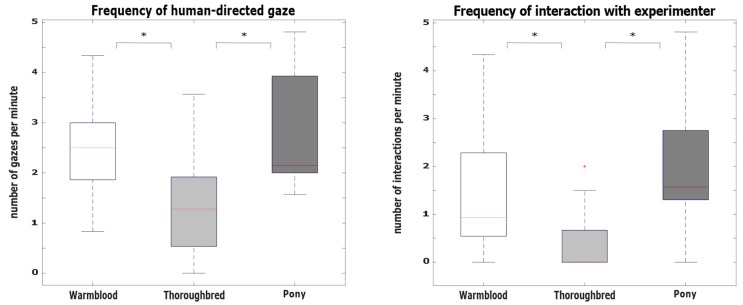
Behavioral differences between breeds. (**a**) Differences in frequency of human-directed gaze between breeds (*H*(2) = 12.00, *p* = 0.002, *d* = 1.10). Thoroughbreds displayed fewer gazes towards humans compared to ponies and warmbloods. (**b**) Differences in frequency of physical interactions with human between breeds (*H*(2) = 13.07, *p* < 0.001, *d* = 1.18). Thoroughbreds interacted less frequently with human compared to ponies and warmbloods.

**Table 1 animals-10-00164-t001:** Sample characteristics.

	Full Sample *n* = 46	Positive Condition *n* = 22	Negative Condition *n* = 24
Age	11.11 ± 7.2 (1–26)	11.32 ± 6.8 (2–23)	10.92 ± 7.8 (1–26)
Male	58.7% (28)	68.2% (15)	54.2% (13)
Female	39.1% (18)	31.8% (7)	45.8% (11)
Warmbloods	43.5% (20)	40.9% (9)	45.8% (11)
Thoroughbreds	41.3% (19)	45.5% (10)	37.5% (9)
Ponies	15.2% (7)	13.6% (3)	16.7% (4)

Values represent for age: mean ± SD (range) and for sex/breed: percent (*n*).

**Table 2 animals-10-00164-t002:** Ethogram of all behaviors analyzed.

**Gaze**		
Gaze human	Frequency	Number of horse’s head orientations towards human
Gaze object	Frequency	Number of horse’s head orientations towards object
**Interaction**		
Interaction human	Frequency	Number of horse’s physical contacts with human
Interaction object	Frequency	Number of horse’s physical contacts with object
**Position**		
Behind experimenter	Duration	Total time of horse’s position between the experimenter and close to the door, farthest from object
Abreast to experimenter	Duration	Total time of horse’s position next to experimenter between door and object
In front of experimenter	Duration	Total time of horse’s position between the experimenter and the object

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
