# Peer review of "Social Referencing in the Domestic Horse"

_animals, 2020, doi:10.3390/ani10010164_

Round 1

Reviewer 1 Report

An interesting study, which aimed to investigate social referencing in horses.  This is a topic which will certainly be of interest to readers of the journal. 

The paper does, however, need some revision, as well as attention to the English.

First, the authors should clarify what they mean by "message".  How were "emotional messages" (by the experimenter) defined?  And how did the handler learn to make a "friendly" face?  These are important details. What is the difference between an "excited" vs an "anxious" face, and what might be the salient differences to the horse?

Second, I would like to see more clarity regarding breed choice (and the associated hypothesis).  "Pony" is not defined - yet there is considerable difference between pony breeds in - for example - reactivity.  I think there should be more information about selection of subjects by breed.  Surely one could argue that warmbloods, too, are selected for performance? Why assume a priori that warmbloods were more likely to be attuned to human gestures than thoroughbreds?

Third, although the paper focuses on horses' responses to different human facial expressions,  surely other factors in human responses could be involved? It is quite possible that when the experimenter tried to use an anxious expression that his/her heartrate went up - to which horses would respond.

Fourth, I was a little surprised that gazing was the primary measure used.  If horses are looking at a novel  object,  I would expect that other responses were likely.  Why choose gazing as a measure?

Finally, the discussion section is too long, especially the section on breed differences.  This could be summarised much more succinctly.

There are several minor points about use of English.  For example, in the brief summary, last line, the word "affect" should be used, not "effect".  There are a few other, quite minor, ones, which should be checked by a native speaker.

Author Response

An interesting study, which aimed to investigate social referencing in horses. This is a topic which will certainly be of interest to readers of the journal. 

Authors’ response: Thanks for the positive feedback!

The paper does, however, need some revision, as well as attention to the English.

First, the authors should clarify what they mean by "message".  How were "emotional messages" (by the experimenter) defined?  And how did the handler learn to make a "friendly" face?  These are important details. What is the difference between an "excited" vs an "anxious" face, and what might be the salient differences to the horse?

Authors’ response: We rephrased the term “message” to “expression”, as message implies an a priori communication between human and horse (which we were up to investigate). The experimenter was instructed to act happy or anxious, with accompanying body tension, pitch of voice, and facial expression. We are aware that the emotional expressions of this experimenter might not be identical (or even similar) to the expression of happiness and anxiety of other humans. We are also aware that distinct expressions cannot be exactly repeated (and thus presented) in an identical fashion to different subjects. To control for this, future studies might use playbacks and video clips of humans displaying these emotions. As we wanted to implement a naturalistic setup, we opted for live stimuli. We hope that our study might pave the way for future experiments that will investigate which modality of emotional expressions (acoustic, visual, etc) has the strongest impact on horse behaviour.

Second, I would like to see more clarity regarding breed choice (and the associated hypothesis).  "Pony" is not defined - yet there is considerable difference between pony breeds in - for example - reactivity.  I think there should be more information about selection of subjects by breed. Surely one could argue that warmbloods, too, are selected for performance? Why assume a priori that warmbloods were more likely to be attuned to human gestures than thoroughbreds?

Authors’ response: We thank the reviewer for this important comment. As we recruited subjects via announcement in the horse owners changing room (where the owners entered their horses name, age, sex, and breed in an appended list), we got information about the subjects’ breed. We further classified the varying breeds into the categories pony, thoroughbred, and warmblood to get sufficient group sizes for statistical analysis. We apologize that we did not clarify the definition of these categories and added this information into the participants section of the revised manuscript.

E.g., we identified seven ponies on this list, including three Icelandic Horses, two Haflinger, one German Riding Pony, and one Fell Pony Mix. However, as some breeds cannot sufficiently be defined as horse or pony (there is a controversy about the Icelandic Horse, for instance), we used in addition the official shoulder height criteria of the International Federation of Equestrian Sports (FEI) for ponies, which is below 1.48 m shoulder height (see FEI Veterinary Regulations 2019, https://inside.fei.org/fei/regulations/veterinary). All seven subjects classified as ponies in this study met the FEI shoulder height criterium for pony. We agree that there exist considerable differences in reactivity and breeding objective between these breeds classified as ponies. Given the low number of animals and the variety of breeds, we defined these cut-off points for statistical analysis. We included this issue in the limitation section.

Lastly, the reviewer is right, warmbloods have been selected for sports performance extensively and are prominent in international sports competitions, especially jumping and dressage. However, in addition to performance, warmbloods also have been strongly selected for temperament [3,4]. Further, a study showed that the percentage of thoroughbred blood lines in warmblood stallions was positively associated with more temperament [1], indicating that, although both warmbloods and thoroughbreds are breed for performance (in different disciplines), warmblood horses were also selected for behavioral characteristics. This might be explained by the fact that warmblood horses are also bred to a large extent for amateur and recreational riders, who prefer ease of handling [2]. In line with these studies, we hypothesized to find behavioral differences between these breeds. We included additional information in the introduction section to clarify how we derived this hypothesis.

References:

BudzyĹ„ska, M., Kamieniak, J., Marciniak, B., & SoĹ‚tys, L. (2018). Relationships between thoroughbreds’ contribution in the pedigree and the level of fearfulness and performance in warmblood stallions. Acta Veterinaria, 68(3), 288–300. Górecka-Bruzda, A., Chruszczewski, M. H., Jaworski, Z., Golonka, M., Jezierski, T., DĹ‚ugosz, B., & Pieszka, M. (2011). Looking for an ideal horse: rider preferences. Anthrozoös, 24(4), 379-392. Koenen, E. P. C., Aldridge, L. I., & Philipsson, J. (2004). An overview of breeding objectives for warmblood sport horses. Livestock Production Science, 88(1-2), 77-84. Rothmann, J., Christensen, O. F., Søndergaard, E., & Ladewig, J. (2014). A note on the heritability of reactivity assessed at field tests for Danish Warmblood Horses. Journal of Equine Veterinary Science, 34(2), 341-343.

It now reads:

“As we recruited subjects of varying breeds, we further classified them into the category pony, thoroughbred, and warmblood according to the following definition: Pony included all breeds with an adult shoulder height of below 1,48 m, determined in accordance with the standard pony definition of the International Federation of Equestrian Sports (FEI, Veterinary Regulations 2019). The category thoroughbred defined all horses with >75 % thoroughbred blood lines in their pedigree (based on owners’ reports). Warmblood was defined as horses with <75 thoroughbred blood lines in their pedigrees (based on owners’ reports).” Lines 160-166

Third, although the paper focuses on horses' responses to different human facial expressions, surely other factors in human responses could be involved? It is quite possible that when the experimenter tried to use an anxious expression that his/her heartrate went up - to which horses would respond.

Authors’ response: We thank the reviewer for this important question. We used human facial expressions and vocalizations, but the reviewer is right, the body posture was adjusted as well. We added this information in the procedure section of the revised manuscript. We further included in the limitation section, that horses’ responses might be influenced by other factors such as heart rate or tone pitch.

It reads:

“Additionally, the horses might have relied on other cues, such as heart rate or tone pitch, that could have been directly affected by the provided human emotional expression. Future studies could, for instance, monitor heart rate of the experimenter and the test subject to gain more precise insights into, which human bodily cues most strongly affect horses’ behavior.” Lines 406-409

Fourth, I was a little surprised that gazing was the primary measure used. If horses are looking at a novel object, I would expect that other responses were likely. Why choose gazing as a measure?

Authors’ response: We used gazing towards the objects as well as toward the experimenter as one main outcome variable. In addition, we also measured physical interactions with the human and the object, as well as the time spent behind or in front of the experimenter. Gaze has been used as a primary measure in previous studies on social referencing in children [1,2,8], dogs, and cats [4,5,6,9]. Gaze has also been used as primary measure in studies with horses on emotion recognition and referential communication [3,7]. As we wanted to ensure comparability with previous studies, we choose gaze, in addition to physical interactions and position in the arena, as one of our main outcome variables.

References:

Hoehl, S., Wiese, L., & Striano, T. (2008). Young infants' neural processing of objects is affected by eye gaze direction and emotional expression. PLoS ONE, 3(6). Klinnert, M. D., Emde, R. N., Butterfield, P., & Campos, J. J. (1986). Social referencing: The infant's use of emotional signals from a friendly adult with mother present. Developmental Psychology, 22(4), 427- Malavasi, R., & Huber, L. (2016). Evidence of heterospecific referential communication from domestic horses (Equus caballus) to humans. Animal Cognition, 19(5), 899–909. Merola, I., Prato-Previde, E., & Marshall-Pescini, S. (2012). Social referencing in dog-owner dyads? Animal Cognition, 15(2), 175–185. Merola, I., Prato-Previde, E., Lazzaroni, M., & Marshall-Pescini, S. (2014). Dogs’ comprehension of referential emotional expressions: Familiar people and familiar emotions are easier. Animal Cognition, 17(2), 373–385. Merola, I., Lazzaroni, M., Marshall-Pescini, S., & Prato-Previde, E. (2015). Social referencing and cat–human communication. Animal Cognition, 18(3), 639–648. Smith, A. V., Proops, L., Grounds, K., Wathan, J., & McComb, K. (2016). Functionally relevant responses to human facial expressions of emotion in the domestic horse (Equus caballus). Biology Letters, 12(4). Walden, T. A., & Ogan, T. A. (1988). The development of social referencing. Child Development, 59(5), 1230–1240. Yong, M. H., & Ruffman, T. (2015). Is that fear? Domestic dogs’ use of social referencing signals from an unfamiliar person. Behavioural Processes, 110, 74–81.

Finally, the discussion section is too long, especially the section on breed differences. This could be summarised much more succinctly.

Authors’ response: We thank the reviewer for this comment. As we had two main hypotheses (differences between conditions and between breeds), we treat both analyses as equally important. As the discussion section about breed differences is already much shorter than the discussion section about differences between conditions, we would like to keep the amount of information. Breed differences in horses’ behavior have been studied rarely but have important implication for daily handling and animal welfare. We also want to inform recreational riders to take breeds’ behavioral differences into consideration when purchasing a horse. However, we removed additional information about genetic influences to shorten the discussion section.

There are several minor points about use of English.  For example, in the brief summary, last line, the word "affect" should be used, not "effect".  There are a few other, quite minor, ones, which should be checked by a native speaker.

Authors’ response: Thanks for this suggestion and pointing out the grammar errors. A native speaker has been reviewing the revised version.

Reviewer 2 Report

The authors of this manuscript do a nice job explaining the relevance of social referencing, its prevalence in other domestic species and the fact that we don’t know if domestic horses engage in the behavior. 

They have designed what I think to be a rigorous experiment to test this behavior in domestic horses.  I do think it would have been great if they could have exposed each animal to the different conditions and tested differences in individuals’ responses as this would have controlled for variations in individual response to novel objects, but I also recognize that one exposure to such stimuli can very well affect subsequent behavior. 

I am a bit concerned about the fact that age and sex differences were not considered—the authors list that these differences were “counterbalanced” between groups and so were not considered—however, I am not entirely convinced that the balance means that, on average, older animals didn’t respond differently than younger animals; similarly for males and females.  As these factors were not considered, we have no way of knowing if there are differences on this level.  Such information could be important for domestic horse owners. 

In addition, I was unable to access any of the 3 figures they refer to in the manuscript.  I triple checked the submission and the figures simply are not there—I emailed the editor in charge of this manuscript hoping to gain access to the figures, but did not receive a reply.  It is necessary for me to see the figures if I am to give a thorough review of this manuscript.

Finally, although the manuscript is well-written, there were several grammatical errors throughout.  For example, the use of the term “wherefore” is incorrect.  Wherefore directly translated means “for what reason”.  I believe the authors mean to use the term “therefore” throughout the paper. I suggest that the authors have the paper reviewed by a native English speaker before resubmission. 

Author Response

The authors of this manuscript do a nice job explaining the relevance of social referencing, its prevalence in other domestic species and the fact that we don’t know if domestic horses engage in the behavior. 

Authors’ response: Thanks for the positive feedback!

They have designed what I think to be a rigorous experiment to test this behavior in domestic horses. I do think it would have been great if they could have exposed each animal to the different conditions and tested differences in individuals’ responses as this would have controlled for variations in individual response to novel objects, but I also recognize that one exposure to such stimuli can very well affect subsequent behavior. 

Authors’ response: Thanks for this valuable suggestion. We wanted to test naïve subjects. As the reviewer has pointed out, repeated exposure might lead to habituation. We agree that future studies should investigate within-subject differences in social referencing paradigms, e.g. by introducing two novel objects simultaneously, one approached positively and one negatively. This has been successfully done in children [2] and in dogs [1]. However, as our study is the first investigating social referencing in horses, we wanted to examine horses general social referencing abilities in a simple setup. In future studies we might plan to further expand the experimental setup and might apply a within-subject design.

References:

Buttelmann, D., & Tomasello, M. (2013). Can domestic dogs (Canis familiaris) use referential emotional expressions to locate hidden food? Animal Cognition, 16(1), 137–145. Walden, T. A., & Ogan, T. A. (1988). The development of social referencing. Child Development, 59(5), 1230–1240.

It reads:

“Lastly, some studies suggest sex differences in horses’ behavior during training or handling [46,47]. We therefore examined potential sex differences in horses’ behavior during the social referencing paradigm.” Lines 147-149.

 “As age did not correlate with any behavioral measure and was counterbalanced between groups, age was not included in the analysis” Lines 214-215

 “Sex differences in behavioral measures were examined using Mann-Whitney U-tests.” Line 223

 “Gazing behavior: Frequency of human-directed gaze (U = 244, p = .848) as well as frequency of object-directed gaze (U = 237, p = .735) did not differ between sexes. Physical interactions: Frequency of interactions with the human (U = 219, p = .437) as well as frequency of interactions with the object (U = 208, p = .170) did not differ between sexes. Relative position to the experimenter: The analysis of subjects’ position in the arena showed no differences between sexes for the position next to the experimenter (U = 191, p = .162). However, the time horses stood behind the experimenter differed between sexes (U = 148, p = .017, d = .80), indicating that female horses stayed longer behind the experimenter (percentage of total trial time: Mdn = .97) compared to male horses (percentage of total trial time: Mdn = .35). Further, the time horses stood in front of the experimenter also differed between sexes (U = 146, p = .011, d = .81), indicating that male horses stayed longer in the area between the experimenter and the object (percentage of total trial time: Mdn = .29) compared to female horses (percentage of total trial time: Mdn = .00).” Lines 271-283

 “Behavioral differences between sexes. Finally, we examined potential sex differences in behavioral variables. We found that sexes differed in their relative position to the experimenter, namely, female horses spent more time behind the experimenter, whereas male horses stayed longer in the area between the experimenter and the object. These results are in line with previous research: mares have been found to be more suspicious and anxious than geldings [63], whereas geldings might be more easily desensitized during training than mares [46]. Mares have also been found to be less playful and curious than geldings [58]. Further, compared to mares, stallions physiological stress responses were less pronounced [64]. However, other studies did not find [24,65-67] or found opposing sex-related differences in horses’ behavior [47]. Potential sex-differences in horses’ behavioral patterns might be explained by male and female sexual hormones that are influencing the endocrine system [68]. E.g., mares’ exploration of novel objects was influenced by its oestrous-cycle stage [68]. Future studies should assess horses’ sex steroids in social referencing paradigms to further interpret behavioral differences between the sexes.” Lines 369-381

I am a bit concerned about the fact that age and sex differences were not considered—the authors list that these differences were “counterbalanced” between groups and so were not considered—however, I am not entirely convinced that the balance means that, on average, older animals didn’t respond differently than younger animals; similarly for males and females.  As these factors were not considered, we have no way of knowing if there are differences on this level.  Such information could be important for domestic horse owners. 

Authors’ response: We thank the reviewer for this important question. We did not include age or sex into the analyses as both factors were not part of our hypotheses. However, we agree with the reviewer that these factor might have important implications and we did further analyses. For age, we did not find a correlation with any behavioral measure independent of condition but there was also no correlation with the behavioral measures in each condition. For sex, we did find differences in horses’ position in the arena. We included the results in the revised manuscript and discussed them.

In addition, I was unable to access any of the 3 figures they refer to in the manuscript.  I triple checked the submission and the figures simply are not there—I emailed the editor in charge of this manuscript hoping to gain access to the figures, but did not receive a reply.  It is necessary for me to see the figures if I am to give a thorough review of this manuscript.

Authors’ response: We apologise for this error. It seems something went wrong with the upload of the figures. In the revised manuscript, we inserted figures and tables into the main file.

Finally, although the manuscript is well-written, there were several grammatical errors throughout.  For example, the use of the term “wherefore” is incorrect.  Wherefore directly translated means “for what reason”.  I believe the authors mean to use the term “therefore” throughout the paper. I suggest that the authors have the paper reviewed by a native English speaker before resubmission. 

Authors’ response: Thanks for this suggestion and pointing out the grammar errors. A native speaker has been reviewing the revised version.

Round 2

Reviewer 2 Report

I think the authors did a great job revising this manuscript—I was happy to see that they explored the potential effects of age and sex on individual response to the novel object.  I think the additional information could be of use to horse owners/handlers.  In addition, I thought their additions to the methods was helpful.

I did find a few minor corrections that ought to be made (see below) but otherwise, I think the revision is great.  I genuinely enjoyed reviewing this manuscript and learning more about the complex connections between domestic horses and humans.

Line 348—change breed to “bred”

Line 366—change male to “males”

Line 367—change female to “females”

Line 374—change E.g. to “For example,”

Line 375—change its to “their” (reference to mares’ needs to be plural)

Line 413—delete one of the periods following “investigated”.

Lines 418-420—I noticed that the English spelling of behavior was used here—elsewhere, the US spelling is used.  I’m not sure which spelling is preferred, but perhaps a re-read of the entire manuscript is appropriate to ensure that the spelling is consistent.

Line 425—change influence to “influences” (reference to the emotional expression— even of several caregivers— needs to be singular)

Author Response

Authors’ response: We thank the reviewer for the supportive and encouraging comments and for improving our manuscript. We have corrected each line in the revised manuscript.